# Modulation of Exercise-Induced Muscle Damage, Inflammation, and Oxidative Markers by Curcumin Supplementation in a Physically Active Population: A Systematic Review

**DOI:** 10.3390/nu12020501

**Published:** 2020-02-15

**Authors:** Diego Fernández-Lázaro, Juan Mielgo-Ayuso, Jesús Seco Calvo, Alfredo Córdova Martínez, Alberto Caballero García, Cesar I. Fernandez-Lazaro

**Affiliations:** 1Department of Cellular Biology, Histology and Pharmacology, Faculty of Health Sciences, University of Valladolid, Campus of Soria, 42003 Soria, Spain; fernandezlazaro@usal.es; 2Department of Biochemistry and Physiology, Faculty of Health Sciences, University of Valladolid, Campus of Soria, 42003 Soria, Spain; juanfrancisco.mielgo@uva.es (J.M.-A.); a.cordova@bio.uva.es (A.C.M.); 3Institute of Biomedicine (IBIOMED), Physiotherapy Department, University of Leon, Campus of Vegazana, 24071 Leon, Spain; dr.seco.jesus@gmail.com; 4Department of Anatomy and Radiology, Faculty of Health Sciences, University of Valladolid, Campus of Soria, 42003 Soria, Spain; albcab@ah.uva.es; 5Department of Preventive Medicine and Public Health, School of Medicine, University of Navarra, IdiSNA, 31008 Pamplona, Spain

**Keywords:** natural polyphenols, curcumin, muscle-damaging exercise, anti-inflammatory, antioxidants, physical activity

## Abstract

Physical activity, particularly high-intensity eccentric muscle contractions, produces exercise-induced muscle damage (EIMD). The breakdown of muscle fibers and the consequent inflammatory responses derived from EIMD affect exercise performance. Curcumin, a natural polyphenol extracted from turmeric, has been shown to have mainly antioxidant and also anti-inflammatory properties. This effect of curcumin could improve EIMD and exercise performance. The main objective of this systematic review was to critically evaluate the effectiveness of curcumin supplementation on EIMD and inflammatory and oxidative markers in a physically active population. A structured search was carried out following Preferred Reporting Items for Systematic Review and Meta-Analyses (PRISMA) guidelines in the databases SCOPUS, Web of Science (WOS), and Medline (PubMed) from inception to October 2019. The search included original articles with randomized controlled crossover or parallel design in which the intake of curcumin administered before and/or after exercise was compared with an identical placebo situation. No filters were applied to the type of physical exercise performed, the sex or the age of the participants. Of the 301 articles identified in the search, 11 met the established criteria and were included in this systematic review. The methodological quality of the studies was assessed using the McMaster Critical Review Form. The use of curcumin reduces the subjective perception of the intensity of muscle pain; reduces muscle damage through the decrease of creatine kinase (CK); increases muscle performance; has an anti-inflammatory effect by modulating the pro-inflammatory cytokines, such as TNF-α, IL-6, and IL-8; and may have a slight antioxidant effect. In summary, the administration of curcumin at a dose between 150–1500 mg/day before and during exercise, and up until 72 h’ post-exercise, improved performance by reducing EIMD and modulating the inflammation caused by physical activity. In addition, humans appear to be able to tolerate high doses of curcumin without significant side-effects.

## 1. Introduction

Physical activity, particularly high-intensity eccentric muscle contractions, induces exercise-induced muscle damage (EIMD) [1,2]. EIMD leads to the onset of an inflammatory response that is associated with a decrease in the ability to generate muscle strength, decreased range of motion (ROM), localized swelling, delayed onset muscle soreness (DOMS), and increased muscle proteins in the blood (creatine kinase (CK), lactate dehydrogenase (LDH), and myoglobin (Mb)) [3]. In addition, EIMD triggers inflammatory responses that result in elevations of inflammation markers such as C-reactive protein (CRP) and some inflammatory interleukins (IL-1, IL-6, tumor necrosis factor (TNF-α)) [4]. Similarly, it promotes the production of transcription factors such as nuclear factor kB (NF kB) through the production of reactive oxygen species (ROS) [5].

On the other hand, research indicates that oxidative stress (OS) is evident following EIMD by an increase in ROS [6]. In this sense, endogenous antioxidants may also be up-regulated via exercise, which stimulates an acute OS and inflammatory response [7]. Therefore, inflammatory processes are always linked to OS and must be analyzed and controlled together, because both are directly involved in EIMD [8]. One way to prevent and minimize the effects of OS and the inflammatory process, and attenuate EIMD [9], could be an oral intake of anti-inflammatory or antioxidant supplementation. A natural product that can be used, with potential antioxidant and anti-inflammatory effects, is curcumin (1,7-bis (4-hydroxy-3-methoxyphenyl) 1,6-heptadiene-3,5-dione), which is the main natural bioactive polyphenol of the spice herb turmeric (2%–5% by weight). Curcumin is a highly pleiotropic molecule that interacts with multiple anti-inflammatory and antioxidant pathways [10,11]. The United States Food and Drug Administration (FDA) has listed curcumin as GRAS (generally recognized as safe), and curcumin-containing supplements have been approved for human ingestion [12]. In this way, curcumin used as a pharmaceutical preparation has been shown to be safe, even at high doses. However, it has been shown to cause some gastric irritation in humans, hepatotoxicity in mice, and at high doses, hepatotoxicity in rats. Humans appear to be able to tolerate high doses of curcumin without significant side-effects. This may be because of differences in metabolism of curcumin in humans as compared to susceptible species such as rats. However, when used as a spice, because of its high water content, it could be attacked by aflatoxin-producing fungi, causing kidney, lung, or liver toxicity. It must be taken into account that curcumin as a spice is produced in tropical countries (warm and humid) that favor the growth of fungi [11,12].

Curcumin supplementation could be beneficial for attenuating EIMD given that curcumin has been shown to potentially help alleviate exercise performance decrements following intense and challenging exercise, as a result of membrane protective effects, antioxidants response, and anti-inflammatory action [13,14]. The anti-inflammatory properties attributed to curcumin are due to its ability to inhibit the nuclear factor kappa (NF-κB), which may be a muscle protective and regeneration agent and plays an important role in controlling physiological mechanisms of inflammation and protein breakdown [15]. Curcumin is capable of blocking the activation of TNF-α-dependent NF-κB and the activation pathway induced by ROS [16,17,18]. Likewise, curcumin could have a low regulatory effect on the expression of the COX-2 enzyme and inhibit pro-inflammatory enzyme 5-LOX (lipoxygenase-5) expression in the leukotriene-producing metabolic pathway [19], as well as the intercellular adhesion molecule 1 (ICAM-1) and vascular cell adhesion molecule 1 (VCAM-1)—a crucial step in the inflammatory response—and decrease inducible nitric oxide synthase (iNOS), which is directly responsible for inflammatory damage by blocking of the cytokines responsible for its activation [19,20]. In this way, curcumin induces the negative regulation of pro-inflammatory interleukins (IL-1, IL-2, IL-6, IL-8, and IL-12), inflammatory cytokines, such as TNF-α and monocyte-1 chemotherapeutic protein (MCP-1), through inhibition of the transcription signaling pathway (JAK/STAT) [11]. In addition, the overexpression of Bcl-2 or Bcl-X L protects cells from apoptosis that counteracts proapoptotic and proinflammatory attacks and restores the anti-inflammatory physiological phenotype [18,21]; curcumin controls the response to thermal shock for attenuated muscle damage [22] and biomarkers of muscle damage, such as CK [11].

In line with this, some studies have shown the effects of curcumin supplementation on OS, inflammation, EIMD, and sport performance with controversial results. These inconsistent results may be due, in part, to differences in doses, timing of supplementation, timing of exercise, exercise model, and experimental design between studies [23]. Previous research has indicated that curcumin may have antioxidant, anti-inflammatory, and analgesic effects on DOMS [24]. Furthermore, similar effects of curcumin have been described by Tanabe et al. [1], who found that ingestion before exercise could attenuate acute inflammation, and after exercise, it could attenuate muscle damage and facilitate faster recovery. Drobnic et al. [5] reported a reduction in muscular trauma with a moderate reduction in pain with curcumin supplementation. In contrast, Sciberras et al. [4] did not reveal any statistical difference between intervention with curcumin and placebo in levels of IL-6 and IL-10. In addition, markers of oxidative stress were only slightly increased after exercise in both groups, which does not allow a comparison of the effects of curcumin versus placebo [5], and there were no differences in terms of changes in maximal voluntary contraction (MVC) and serum CK activity [25]. Finally, not all observed changes in performance and soreness after exercise in humans [10] have been reproduced on the mouse model [26].

However, it is necessary to clarify the useful doses, timing (before or after exercise), duration of treatment, and the effects of curcumin on OS, inflammation, and EIMD. Therefore, the purpose of this study was to critically evaluate the effectiveness of curcumin supplementation on EIMD, inflammatory, and oxidative markers in a physically active population. In addition, the study shows the effective doses, timing, and duration of treatment for optimal application.

## 2. Material and Methods

### 2.1. Search Strategy

The present article is a systematic review focusing on the effect of curcumin supplementation on muscle pain, muscle function, muscle enzyme activity, inflammatory markers, and antioxidant effect and was conducted following the Preferred Reporting Elements for Systematic Reviews and Meta-analysis (PRISMA) guidelines [27] and the PICOS question model for the definition of inclusion criteria: P (population); “healthy exercise practitioners”, I (intervention); “supplementation with curcumin”, C (comparison); “same conditions with placebo or control group”, O (outcomes); “muscle pain, inflammation and/or muscle damage serum markers and antioxidant effect”, S (study design); “double- or single-blind design and randomized parallel or crossed”.

A structured search was carried out in the databases SCOPUS, Medline (PubMed), and Web of Science (WOS), which includes other databases such as BCI, BIOSIS, CCC, DIIDW, INSPEC, KJD, MEDLINE, RSCI, SCIELO, all of which are high-quality databases which guarantee good bibliographic support. The search covered a time span from March 2015—when Nicol et al. [28] suggested the use of oral curcumin is likely to reduce pain associated with delayed onset muscle soreness (DOMS) with some evidence for enhanced recovery of muscle performance—to November 2019. Search terms were a mix of Medical Subject Headings (MeSH) and free words for key concepts related to curcumin, muscle, exercise, inflammation, and recovery, as follows: (“curcumin” OR “curcuminoids” OR “curcuma longa” OR “turmeric”) AND (“muscle damage” OR “delay onset muscle soreness” OR “DOMS” OR “inflammation” OR “inflammatory” OR “inflammatory markers” OR “oxidative stress”) AND (“exercise” OR “physical activity” OR “sports”). Through this search, relevant articles in the field were obtained applying the snowball strategy. All titles and abstracts from the search were cross-referenced to identify duplicates and any potential missing studies. Titles and abstracts were then screened for a subsequent full-text review. The search for published studies was independently performed by two authors (DFL and CIFL) and disagreements about physical parameters were resolved through discussion.

### 2.2. Selection of Articles: Inclusion and Exclusion Criteria

For the articles obtained in the search, the following inclusion criteria were applied to select studies: articles (I) depicting a well-designed experiment that included the ingestion of a dose of curcumin, or a curcumin-containing product, before and/or during exercise in humans; (II) with an identical experimental situation with or without the ingestion of a placebo; (III) with a double- or single-blind design and randomized parallel or crossed design; (V) with clear information on the administration of curcumin (dose of curcumin per kg body mass and/or absolute dose of curcumin on body mass, time of curcumin intake before or after exercise, duration of treatment); (VI) in which curcumin was administered in the form of a beverage, gum, or pills; (VII) in which one of the measured variables was changes in muscle pain, muscle function, muscle enzyme activity, inflammatory markers, or antioxidant effect; (VIII) in which the languages were restricted to English, German, French, Italian, Spanish, and Portuguese. The following exclusion criteria were applied: (I) animal studies, (II) uncontrolled trials, (III) studies using non-standardized turmeric extracts or extracts of unknown curcuminoid content, and (IV) studies performed on subjects with a prior condition of musculoskeletal injury or pain. There were no filters applied to the individuals’ fitness level, sex, or age to increase the power of the analysis.

The methodological quality of the articles, evaluated using McMaster’s Critical Review Form [29], scored between 12 and 15 points, representing a minimum methodological quality of 75% and a maximum of 93.8%. Of the 11 studies, 7 achieved a “Very Good” quality, 2 a “Good” quality, and 2 studies an “excellent” quality. No study was excluded because it did not reach the minimum quality threshold. Table 1 details the results of the criteria evaluated, where the main deficiencies found in methodological quality are associated with items 5 and 12 of the questionnaire, and comprises a detailed justification of the size of the study and a discussion of relevance of the results to clinical practice, respectively. The objective of this evaluation was to determine the existing methodological limitations in each of the studies and to allow the quality of the results to be comparable between the different study designs.

Once the inclusion/exclusion criteria were applied to each study including authors, year of publication, study design, curcumin administration (dose and timing), sample size, and characteristics of the participants (fitness level and sex), and final outcomes of the interventions were extracted independently by two authors (DFL and CIFL) using a spreadsheet (Microsoft Inc, Seattle, WA, USA). Subsequently, disagreements were resolved through discussion until a consensus was reached.

## 3. Results

### 3.1. Selection of Studies

The literature search provided a total of 301 articles related to the select descriptors, but only 11 articles met all the inclusion/exclusion criteria (Figure 1). The number of the articles to which each exclusion criterion was applied were as follows: 94 papers were removed because they were duplicates. After the elimination of duplicate articles, 207 articles were selected for examination by title and abstract, of which 150 were excluded as non-intervention studies and 38 as unrelated to the search topic. The full texts of the remaining 19 publications were evaluated according to the inclusion criteria, from which three studies were eliminated because they were conducted in animal populations, four because they used unhealthy subjects, and one because they did not measure any of the variables included in this study.

### 3.2. Characteristics of the Studies

The participants ‘samples (*n* = 237) included individuals of both genders (187 men and 50 women), where 10 were elite athletes, 131 were moderately active people, and 96 were people who were asked not to undergo pre-study training (Table 2). In ten of the eleven articles, some commercial supplement of curcumin in capsules of standardized composition and known bioavailability was used, while Nicol et al. [28] opted for a pharmaceutical preparation of curcumin capsules using a specific composition protocol for research. In addition, Delecroix et al. [31] chose a combination of curcumin plus piperine in the composition. Regarding the daily dose of curcumin, seven studies used doses ranging from 150 to 1500 mg [1,2,4,5,10,24,25,30,32], and two studies tested higher doses of about 5 g [28] and 6 g [31] daily. In nine of the included studies, supplementation was given before and after exercise [1,2,5,10,25,28,30,31,32], Sciberras et al. [4] used curcumin before exercise, and Nakhostin-Roohi et al. [24] supplemented with curcumin after exercise. Finally, treatment duration ranged from one to fifty-six days, with three studies of four days [1,5,31], two of six days [28,30], two of seven days [1,25], two of one day [2,24], one of twenty-eight days [10], and one of fifty-six days [32], respectively.

### 3.3. Outcome Measures

Table 3A–C includes information about the author/s and year of publication; the sample investigated, with details of fitness level, sex, and the number of participants; the study design cites the control group if the study included one; the supplementation protocol that specifies the type of curcumin used, the dose, and the time at which it was administered; the parameters analyzed or main effects on muscle damage; and finally, results or main conclusions.

## 4. Discussion

The main objective of this systematic review was to critically evaluate the effectiveness of curcumin supplementation on EIMD (muscle pain, muscle performance, and muscle enzyme activity) and inflammatory and oxidative markers in a physically active population. The main results indicated that supplementation with 150 and 1500 mg/day of oral curcumin, both before and up until 72 h after exercise, has been shown to be effective on exercise performance, modulated in part by the reduction of EIMD and inflammation caused by physical activity. However, it was difficult to determine the true efficacy of the antioxidant capacity of curcumin. Due to the differently measured outcomes in the studies, the following outcomes were divided into different groups to provide a clearer analysis. The results could be influenced by type of exercise, amount of each supplement, and duration of the intervention. Participant characteristics, such as age, gender, ethnicity, body composition, training level, differences in training, nutrition, and health status, may also have influenced the results.

### 4.1. Curcumin Supplementation

The dose of curcumin administered in interventions ranged from 150 to 6000 mg/day, and therefore, the effects of curcumin in a physically active population should be attributed to this dose range. However, the European Food Safety Authority (EFSA) determined the permitted daily intake to be 3 mg/kg body weight [33]. Thus, two investigations justified a dose of 180 mg/day of curcumin following the EFSA recommendation [1,25]. However, other studies [28,30,32] did not base the choice of dose on the same criteria.

Mc Farlin et al. [30] designed a pilot experiment in which they compared the effect of three doses of curcumin (200, 400, and 1000 mg) on inflammatory serum cytokines in order to avoid the use of animal models. These authors concluded that the optimal dose would be 400 mg/day. Furthermore, Nicol et al. [28] selected a dose of 5000 mg/day of natural curcumin based on a study in mice, which resulted in a dose of less than 5% bioavailable curcumin (<200 mg/day of bioactive curcumin). Finally, Basham et al. [32] selected a dose of 1500 mg (69 mg of curcuminoids). This study implemented a newly enhanced absorption and pharmacokinetics of fresh turmeric derived from curcuminoids in comparison with the standard curcumin from dried rhizomers. Dosing was determined via the manufacturer’s recommendations [34]. However, most studies investigating curcumin supplementation have utilized dosages ranging from as little as 50 mg/day [35] to 2.5 g/day [28], demonstrating efficacy and safety in humans. Further, internal quality control testing was performed by the manufacturer, ensuring safety and authenticity of the supplement.

Thus, while research using curcumin supplements with improved bioavailability (Meriva Curcumin [4,5]; Theracurcumin Theravalues [1,24,25,36]; Longvida [30]; CurcuWIN [10]; CurcuFresh [32] determined the dose at values between 150 and 1500 mg/day, studies using curcumin naturally [28,31] needed higher doses (5000 to 6000 mg/day) to achieve similar bioavailability.

The Korean Food and Drug Safety Administration has declared turmeric safe and tolerable in humans, and long-term studies with curcumin have revealed no toxic or adverse effects. However, in a supradosing range, with higher doses than the studies described in this manuscript of between 8 and 12 g, some subjects experienced mild nausea or diarrhea [37]. One consideration that should be taken into account in supplementation with curcumin in athletes, who are themselves susceptible to iron deficiencies with or without anemia, is the interaction between high doses of curcumin and the alteration of iron metabolism by the chelation of iron and elevation of hepcidin, which could be another potential cause of decreased iron levels [38]. For this reason, piperine increases in importance because it allows a high bioavailability of curcumin with lower doses [39]. In this sense, Delecroix et al. [31] used curcumin (6 g) plus piperine (20 mg) per day. However, it is not possible to determine the degree of absorption of the different formulations of curcumin because the studies do not reveal the plasma concentrations.

We believe that curcumin can be safely used as a modulating therapy for markers of inflammation and exercise-induced muscle damage. However, in spite of the safety of the curcumin dose, it is necessary to develop more precise criteria according to the type of curcumin administered, the duration of treatment, and the type of sport performed, in order to establish an optimal dose and an effective intake time that are capable of attenuating the effects of exercise on inflammatory responses and muscle damage.

### 4.2. Exercise-Induced Muscle Damage (EIMD)

EIMD could affect different muscle dimensions such as muscle pain, muscle performance, and muscle enzyme activity.

#### 4.2.1. Effect on Muscle Pain

Muscle pain can be induced by EIMD or an unaccustomed activity [40] and results in discomfort at the site of the injury and loss, among others, of muscle function and strength; hence, it limits physical function for several days after exercise [41]. The potential effect of curcumin supplementation in reducing muscle pain could be due to the effect it has on suppressing the induction of expression of the isoform COX-2 [25], thus avoiding the production of mediating substances, such as prostaglandin E2 (PGE2), histamine, bradykinin, and serotonin derived from COX-2 that activate nerve endings [19]. The action of curcumin decreasing these mediators, especially PGE2, would provide the attenuation of the phenomenon of long-lasting hyperalgesia that occurs in afferent sensory fibers of type C [42].

In this sense, ten studies [1,2,5,10,24,25,28,30,31,32] evaluate the ability of curcumin to attenuate muscle pain; eight of which do so through the Visual Analog Scale (VAS) [1,2,4,10,24,25,28,32]. Concretely, Nicol et al. [28], described that the intake of 2.5 g of curcumin supplementation in capsules taken 48 h before and 72 h after eccentric exercise caused significant reductions in pain. Aligned with this, in two studies by Tanabe et al. [1,25], a significant reduction in muscle pain was demonstrated by the effect of administering 180 mg of curcumin supplement (90 mg twice daily) in Theracumin-Theravalues capsules, only when administered four [25] and seven [1] days after eccentric contraction exercise. In addition, the curcumin supplementation (1.5 g) resulted in significantly decreased muscle soreness overall (VAS scale 2.88) when compared to the placebo (VAS scale 3.36) (*p* < 0.0120) [32]. The supplementation continued for three days during the follow-up testing sessions; thus, 28 total days of supplementation were implemented. Finally, Nakhostin-Roohi et al. [24] showed that one dose of 150 mg of curcumin supplementation in capsules (Theravalues) taken immediately after exercise significantly reduced muscle pain at 48 and 72 h after eccentric exercise.

However, Jäger et al. [10] showed non-significant improvements in exercise-induced total thigh soreness and indicated that the 200 mg curcumin groups reported 26%, 20%, and 8% less soreness immediately, 24 h, and 48 h after exercise, respectively, as compared the soreness levels that were reported in the PLA and 50 mg curcumin groups; these differences failed to reach statistical significance. The supplementation of curcumin was for eight weeks prior to downhill running protocol VAS. In this sense, Tanabe et al. [2] did not find a significant effect in delayed onset muscle soreness (DOMS) using 150 mg of curcumin or placebo orally before and 12 h after each eccentric exercise.

There are many possible supplementation conditions in terms of dose, frequency, and time points. Nosaka et al. [43] reported that essential amino acid supplementation given both 30 min before and immediately after eccentric exercise did not affect any indicators of muscle damage, but when the supplementation was continued for next four days after exercise, it attenuated increase in muscle soreness and range of motion (ROM). Thus, it is possible that curcumin supplementation has an effect on DOMS if the curcumin is given on recovery days, for a period of at least 3 days, as in [28,32], or for a longer period of time [1,25]. However, a single dose of curcumin (150 mg) immediately after exercise significantly reduced muscle pain [24]. This situation may justify the need for a series of studies to investigate whether curcumin supplementation for recovery immediately after exercise will provide greater effects on muscle damage.

On the other hand, three studies [5,30,31] evaluate the ability of curcumin to attenuate muscle pain using other scales. In particular, Drobnic et al. [5] observed a tendency towards less pain, but no significant improvements, in the lower extremities after 200 mg curcumin supplementation taken 48 h before and 24 h after a downhill race. In addition, Mc Farling et al. [30] investigated subjective quadriceps muscle soreness, finding no significant difference in muscle soreness or activities of daily living soreness between supplementation with curcumin (400 mg; 24 h before and for 72 h after) and placebo. Moreover, Delecroix et al. [31] showed no beneficial effect of curcumin (2 g; 48 h before and for 48 h after) in reducing muscle pain, as evaluated by the Hooper scale and subjective quadriceps muscle soreness.

It is possible that differences in pain perception outcomes in studies that could not establish the same effect are probably due to the subjective perception of the patient’s pain intensity. For this reason, a combination of physiological and psychological factors could play an important role in individual perception, thus potentially improving recovery from training [44]. Therefore, it cannot be ruled out that perception, psychological factors, and placebo effects may have influenced the reported results. We described that the doses of curcumin that showed benefits for muscle pain attenuation were in a wide range (150 mg–2.5 g); however, what might be effective is administration immediately after exercise and/or within at least 72 h after exercise.

#### 4.2.2. Effect on Muscle Performance

Muscle damage caused by mechanical stress during eccentric exercise and subsequent inflammatory responses lead to a deterioration of muscle performance. Thus, changes in maximum voluntary contraction (MVC) force, range of motion (ROM), and isokinetic dynamometry reflect the dimension and time progression of EIMD, and therefore, these parameters can be used as markers of athletic performance [1,2,31]. MVC, isokinetic dynamometry, and ROM are diminished by the activation of NF-κB under the high mechanical stress caused by the overuse of some joints in sport, which generates fragments of the extracellular matrix of the bone or cartilage. These fragments are recognized by receptors of innate immunity, which recognize pattern recognition receptors, called toll-like receptors. Cell activation mediated by this process ends between the activation of NF-κB, which is a stimulator of the secretion of inflammatory cytokines (IL-1, IL-1b, IL-2, IL-15, IL-21, TNF-α), chemokines (CCL-19, CCR-7), and metalloproteases (MMP-13, ADAMSTS-4), all responsible for producing tissue damage [45]. It is probably the action of curcumin that is responsible for minimizing the incapacitating tissue effects, as it is a therapeutic agent that blocks the signaling pathway of NF-κB. In addition, reducing leucocyte adhesion and migration, and as a result, relieving pain and swelling, improves joint mobility and stiffness [16,37]. Three studies conducted by Tanabe et al. [1,2,25] evaluated MVC and ROM. One of them [2] showed that 150 mg of curcumin both before and 12 h after 50 eccentric contractions of the elbow flexors presented a significantly smaller decreasing magnitude on the MVC. However, ROM decreased significantly at all measurement times (24, 48, 72, and 96 h) in the curcumin group, but there were no significant interaction effects for changes between placebo and curcumin. Another study by the same author [1] demonstrated significantly faster recovery of torque MVC and significant improvements in ROM only when 180 mg of curcumin was ingested after exercise [1]. Moreover, Tanabe et al. [25] found that when 180 mg curcumin was ingested during the four days following exercise, ROM improved significantly after three to four post-exercise days, with no relevant changes in MVC for any of the study conditions. Recently, Jäger et al. [10] reported that only 200 mg (CurcuWIN^®^) was effective in preventing the observed decreases in peak extension torque values seen 1 and 24 h after exercise that damaged muscles.

In this way, these studies suggest that the intake of a dose of curcumin (90–200 mg) after exercise may be effective as a muscle performance enhancer by benefiting the recovery process.

#### 4.2.3. Effect on Muscle Enzyme Activity

Intense exercise increases the circulating levels of markers for muscle damage such as lactate dehydrogenase (LDH), CK, myoglobin (Mb), and transaminases (alanine aminotransferase (ALT) and aspartate aminotransferase (AST)). All these parameters are indicative of increased EIMD and OS, which negatively affect athletes because they reduce exercise performance and can also put their health at risk [46].

The effect of curcumin on CK enzyme activity levels after exercise was studied in eight investigations included in this systematic review [1,2,5,24,25,28,30,31]. Five of them [1,2,24,28,30] presented significantly lower maximum CK activity in the curcumin-supplemented group compared to the placebo groups. Moreover, although there were no significant differences, three studies [5,25,31] observed that CK levels tended to increase less in the curcumin group. Potentially, the decrease in CK after curcumin supplementation could be attributed to an antioxidant role by neutralizing oxygen free radicals (ROS) produced during the electron transport chain of oxidative phosphorylation, necessary for energy requirements in physical exercise [8]. Another possible mechanism of CK’s activity reduction may be the inhibition of the production of histamine and prostaglandin by suppressing the positive regulation of COX-2, a pathway involved in vascular permeability [47]. In local areas with inflammation, they reduce the permeability of the membranes, thus reducing the intracellular–intravascular flow of CK. Thus, the limitation of vascular permeability could be the key factor in reducing inflammation and muscle pain [48]. The differences between studies may be due to dose, timing of curcumin supplementation, and intensity of physical activity. In addition, Nakhostin-Roohi et al. [24] showed ALT and AST at significantly reduced levels in curcumin group.

The findings of this study [24] suggest that a 150 mg dose of curcumin, when administered immediately after exercise, may have protective effects on muscle damage by significantly reducing the levels of three markers of circulating muscle damage (CK, AST, and ALT). It is difficult to concretize supplementation with curcumin because there are many possible supplementation conditions in terms of dose, frequency, and time points where curcumin has been shown to be effective in relation to its potential to decrease muscle damage, which is reflected in decreased CK activity in the supplemented groups.

### 4.3. Effect on Inflammatory Markers

Curcumin is currently recognized for its potential anti-inflammatory effect [13,15]. In this sense, inflammation is a physiological process in response to physical exercise. Pro-inflammatory cytokines and chemokines produced by immune cells during the immune response interact with their receptors by activating signaling pathways of the inflammatory response [20]. The possible positive effect of curcumin supplementation on inflammatory response may be due to the modulating action of curcumin on inflammatory signaling cascades. These signaling pathways include the nuclear factor κB pathway (NF-κB), the signal transducer and transcription activator Janus kinase (JAK/STAT), and mitogen activated protein kinase (MAPKs). Curcumin inhibits the activation of NF-κB, suppresses the activation and phosphorylation of JAK/STAT proteins, and inhibits MAPK signaling through its interaction with three major members of this pathway, including JNK, p38, and ERK. Bisdemethoxycurcumin suppresses infiltration, activation, and maturation of leukocytes, and also the production of proinflammatory mediators TNF-α, IL-8, and IL-6 at the site of inflammation. Another effect the potential of curcumin is to act as an immunomodulator by intervening in the suppression of immune responses acquired in T cells, inhibiting the activation, differentiation, and production of cytokines [20].

In the studies analyzed in this systematic review, the pro-inflammatory cytokines TNF-α [1,2,28,30,32], interleukin 6 (IL-6) [2,4,28,30], and IL-8 [1,5,30] were evaluated as markers of inflammation. In relation to TNF-α, Nicol et al. [28] and Tanabe et al. [1,2] did not observe any effect of curcumin supplementation on this inflammation marker. However, McFarlin et al. [30] and Basham et al. [32] reported sustained suppression of TNF-α for up to one day after exercise in the curcumin supplemented group compared to the placebo group. Moreover, TNF-α was significantly lower with curcumin at two days and four days and trended towards being lower with curcumin at three days compared to placebo [30]. It is likely that the effectiveness of curcumin on the plasma decrease of TNF-α is dependent on a minimum dose of 400 mg administered before exercise and for 72 h after exercise (at minimum), which may explain the differences with respect to the studies by Tanabe et al. [1,2]. In addition, commercial formulas of curcumin [30,32] implemented an enhanced absorption and pharmacokinetics as compared to standard preparations of curcumin [28].

On the other hand, curcumin supplementation showed a downward but non-significant trend on IL-6 cytokines derived from exercise practice [2,4,30]. Moreover, Drobnic et al. [5] and Mc Farlin et al. [30] showed that 400 mg of curcumin supplementation (48 h before and 24–72 h after exercise) significantly reduced IL-8 levels. However, Tanabe et al. [1] observed non-significant decreases in IL-8 12 h after exercise, when subjects were supplemented with 180 mg of curcumin for seven days before the eccentric exercise test. The differences in dosage and timing could be the cause of differences in IL-8 regulation. According to these results, it is possible that a minimum of 400 mg of curcumin administered before and for at least 24 h after exercise may be necessary.

In this way, the promoters of IL-6 and IL-8 cytokines possess binding sites for NF-κB, C/EBPβ, and c-Jun [49]. We believe the role of NF-κB inhibition is a therapeutic objective of curcumin in inflammation because of the importance of NF-κB for the regulation of the constitution and expression of IL-6 and IL-8 [49,50]. McFarling et al. [30] concretely observed that significant inhibition of NF-κB could be related to a significant decrease in IL-8 and a downward trend in IL-6. Thus, supplementation with 400 mg of curcumin, two days before and three days after exercise, appears to be effective in attenuating exercise-induced inflammation because of its direct action on NF-κB, which influences the cytokines IL-8 and IL-6.

Other cytokines, such as IL-10 [4,30], with anti-inflammatory properties capable of inhibiting the synthesis of pro-inflammatory cytokines and IL1RA [4], which is a key mediator in the inflammatory response, were evaluated. Lower levels of non-significant IL-10 and IL1RA were reported in the curcumin-administered groups as compared to the placebo group.

### 4.4. Effect on Oxidative Markers

This mechanical stress derived from exercise triggers an inflammatory response and the production of ROS. ROS are able to maintain inflammation and in general a high degree of oxidative stress by stimulating the activation pathways of transcription factors such as nuclear factor-κB (NF-κB), a pro-inflammatory master switch that controls the production of inflammatory energy, and are involved in cell damage [32]. This sustained inflammatory response and the high degree of oxidative stress lead to the accumulation of neutrophils and increased “inflammatory growth medium” production of oxidative enzymes, cytokines, and chemokines [15]. In every organism, there is a protection system formed by antioxidant compounds and enzymes that participate in the transformations of these species. One of them is the enzyme glutathione peroxidase (GPx). This enzyme uses glutathione to reduce peroxides, thereby protecting membranes and other cellular structures from the action of lipid peroxides and free radicals [51]. In addition, catalase (CAT) is one of the enzymes involved in the destruction of hydrogen peroxide generated during cell metabolism [52]. The overproduction of ROS eventually surpasses the body’s antioxidant capacity [15].

Against this background, curcumin could be useful because it has been described as suppressing the activation of NF-κB and the promotion of antioxidant response by the transcription activation of Nrf2, which could neutralize these harmful effects related to ROS [18]. Drobnic et al. [5] observed non-significant increases in GPx and CAT after supplementation with curcumin (200 mg) before and after exercise. Other antioxidant markers, such as reactive metabolites of oxygen serum (d-ROMs) and potential biological antioxidant (BAP) concentrations, were not different between curcumin and placebo trials after supplementation with curcumin (90 mg) before and after exercise [1]. In agreement with this, Basham et al. [32] found that exercise was not different between curcumin and placebo trials in total antioxidant capacity (TAC), and that malondialdehyde is formed by the lipid peroxidation of unsaturated fatty acids and is a marker of oxidative degradation of the cell membrane. It is difficult to determine the efficacy of the antioxidant capacity of curcumin, considering the above results. However, in this study, TAC remained significantly higher in the curcumin group after exercise compared with the levels in the placebo group (*p* < 0.05) [24]. The findings of this study suggest that a 150 mg dose of curcumin may have antioxidant effects. The differences between the studies could be due to the timing of ingestion of curcumin, manifesting only a significant effect when ingested immediately after exercise, as suggested by Nakhostin-Roohi et al. [24].

## 5. Limitations, Strengths, and Future Lines of Research

The main limitations of the present systematic review are related to the low number of studies investigated on this subject and to the fact that most of them had a relatively small number of participants. Because of this, it is essential to take into consideration that the studies analyzed were conducted in populations with different levels of physical activity and using different research protocols, which increases the heterogeneity between studies. In addition, dosage, duration of treatment, and formulation of curcumin were not uniform across investigations, which could affect some of the results, particularly because of the limitation associated with the bioavailability of the supplement. Likewise, in some of the selected studies, specific diet controls were carried out prior to the study, telling the participants not to eat foods containing curcumin (e.g., curry) or to follow any nutritional guidelines to avoid the main polyphenols in the diet. However, in other studies, only the consumption of anti-inflammatory drugs was limited, whilst it is still possible that some habitual foods produce anti-inflammatory effects. Finally, it should be noted that the studies analyzed only evaluated inflammatory markers from blood samples, but none in muscle tissue.

The development of curcuminoids in nanoformulations to improve the pharmacokinetics, bioavailability, and biological activity of curcuminoids is currently under investigation [53]. Further research is needed to determine whether these new formulations might be more effective in treating inflammation and the time course of exercise-induced muscle damage. Another way to improve bioavailability would be through piperine, as demonstrated by Delecroix et al. [31], who included 6 g of curcumin +60 mg of piperine/day. Piperine is a thermonutrient that exerts its thermogenic action on the epithelial cells of the small intestine, increasing the rate of nutrient absorption and therefore increasing its bioavailability [54].

Although the studies used in this review were not conducted exclusively in a competitive sports population, they provide an adequate justification to support broader research that could eventually confer its acceptance as a standard in the recovery of muscle function and inflammatory parameters in the framework of physical activity. Furthermore, a strength of this systematic review is quality control through PRISMA and Mc Master.

## 6. Practical Applications

In general, curcumin supplementation could be used during periods of high demand, tournaments, or competitive events to speed up the recovery of muscle function and counteract the size and progress of symptoms associated with exercise-induced muscle pain [14]. In addition, curcumin has been used in various protocols before and/or after exercise, to decrease inflammation and muscle damage through its ability to modulate the inflammatory response and its antioxidant effect. However, implementing curcumin as an ergogenic aid should be considered in light of the sporting objective, since the persistent use of anti-inflammatory substances that benefit recovery could affect training adaptations [55]. However, maximizing resilience at the expense of training adaptations may be desirable in athletes in competitive seasons.

Since the athlete’s recovery in a period of high training or competition loads is limited to a specific moment, the decision on the supplementation strategy to be implemented should be made considering the physiological effects of the substance. Along these lines, it could be considered that supplementation with curcumin in combination with one or more substances that act on different physiological mechanisms could result in a synergistic effect on the parameters of inflammation and muscle damage in the recovery process.

Finally, curcumin should always be used as a pharmaceutical preparation and not as a spice to avoid the toxic effects of fungical aphlotoxins. In addition, caution is needed with athletes who are sensitive to gastric irritation.

## 7. Conclusions

In summary, the use of curcumin reduces the subjective perception of the intensity of muscle pain. Likewise, curcumin is able to decrease muscle damage through the reduction of muscle CK activity and to increase muscle performance. Moreover, supplementation with curcumin exerts a post-exercise anti-inflammatory effect by modulating the pro-inflammatory cytokines TNF-α, IL-6, and IL-8, and curcumin may have a slight antioxidant effect. The minimum optimal dose to achieve a positive impact would be recommended doses between 150 and 1500 mg/day, when administered before and immediately after exercise, and for 72 h after. Finally, curcumin should only be recommended to athletes who are willing to use ergogenic aids to increase performance, and it should be recommended only on an individual basis to modulate some of the muscle damage and inflammation caused by physical activity. Oral curcumin supplementation has been shown to be effective pre and/or post physical activity.

## Figures and Tables

**Figure 1 nutrients-12-00501-f001:**
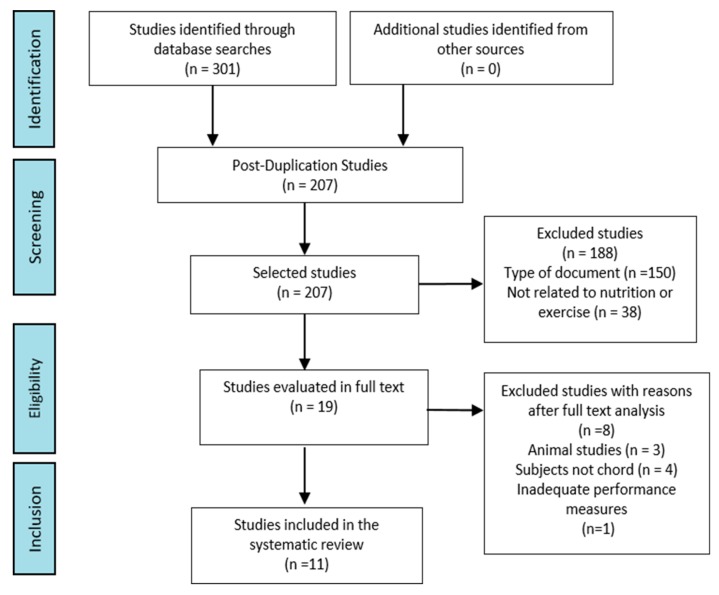
Selection of Studies.

**Table 1 nutrients-12-00501-t001:** Methodological quality of the studies included in the systematic review.

**References**		**Drobnic et al., 2014 [5]**	**Sciberras et al., 2015 [4]**	**Nicol et al., 2015 [28]**	**Tanabe et al., 2015 [2]**	**McFarlin et al., 2016 [30]**	**Nakhostin-Roohi et al., 2016 [24]**	**Delecroix et al., 2017 [31]**	**Tanabe et al., 2018 [1]**	**Tanabe et al., 2019 [25]**	**Jäger et al., 2019 [10]**	**Basham et al., 2019 [32]**	**T_I_**
**ITEMS**	**1**	1	1	1	1	1	1	1	1	1	1	1	10
	**2**	1	1	1	1	1	1	1	1	1	1	1	10
	**3**	1	1	1	1	1	1	1	1	1	1	1	10
	**4**	1	1	1	1	1	1	1	1	1	1	1	10
	**5**	0	1	0	1	0	0	0	1	1	1	1	4
	**6**	0	1	0	1	1	1	1	1	1	1	1	7
	**7**	1	1	1	1	1	1	1	1	1	1	1	10
	**8**	1	1	1	1	1	1	1	1	1	1	1	10
	**9**	1	1	1	1	1	1	1	1	1	1	1	10
	**10**	1	1	0	1	1	1	0	1	1	1	1	7
	**11**	1	1	1	1	1	1	1	1	1	1	1	10
	**12**	1	0	0	1	1	0	0	0	0	0	0	3
	**13**	0	0	1	0	1	1	1	1	1	1	1	6
	**14**	1	1	1	1	1	1	1	1	1	1	1	10
	**15**	1	1	1	1	1	1	0	1	0	1	0	7
	**16**	1	1	1	0	0	0	1	1	1	1	1	6
**T_S_**		13	14	12	14	14	13	12	15	14	15	14	
**%**		81.3	87.5	75	87.5	87.5	81.3	75	93.8	87.5	93.8	87.5	
**MQ**		VG	VG	G	VG	VG	VG	G	E	VG	E	VG	

(T_S_) Total items fulfilled by study. (1) Criterion met; (0) Criterion not met. (T_I_): Total items fulfilled by items. Methodological Quality (MQ): poor (P) ≤8 points; acceptable (A) 9–10 points; good (G) 11–12 points; very good (VG) 13–14 points; excellent (E) ≥15 points.

**Table 2 nutrients-12-00501-t002:** Characteristics of participants and interventions in the studies included in the review.

**Level of Participants**	Elite Athletes	1 Study [31]
	Moderately Active	5 Studies [4,5,10,28,32]
	No Regular Training before the Study	5 Studies [1,2,24,25,30]
**Type of Administration of Curcumin**	Commercially available curcumin supplement	10 studies [1,2,4,5,10,24,25,30,31,32]
	Curcumin capsule made for the study	1 study [28]
**Dosage Used**	150 mg/day	1 study [24]
	180 mg/day (2 doses of 90 mg/day)	2 studies [1,25]
	300 mg /day (2 doses of 150 mg/day)	1 study [2]
	400 mg/day	1 study [30]
	400 mg /day (2 doses of 200 mg/day)	1 study [5]
	500 mg / day	1 study [4]
	5 g/day (2 doses of 2.5 g/day)	1 study [28]
	6 g of curcumin + 60 mg of piperine/day (3 doses of 2 g of curcumin + 20 mg of piperine/day)	1 study [31]
	600 mg/day (3 doses of 200 mg of curcumin)	1 study [10]
	1500 mg/day (3 doses of 500 mg of curcumin)	1 study [32]
**Moment of Supplementation**	Before Exercise	1 study [4]
	Before and After Exercise	9 studies [1,2,5,10,25,28,30,31,32]
	After Exercise	1 study [24]
**Duration of Treatment**	1 day	2 studies [2,24]
	4 days	3 studies [4,5,31]
	6 days	2 studies [28,30]
	7 days	2 studies [1,25]
	56 days	1 study [10]
	28 days	1 study [32]

**Table 3 nutrients-12-00501-t003:** Summary of studies included in this systematic review.

**A. Summary of Studies Included in This Systematic Review.**					
**Author/s—Year**	**Study Design**	**Population**	**Intervention**	**Analyzed Results**	**Main Conclusions**
Drobnic et al., 2014 [5]	Randomized controlled trial single-blind	20 moderately active men (38.1 ± 11.1 years and 32.7 ± 12.3 years)	200 mg curcumin capsules (Phytosome Meriva) twice a day 48 h before exercise and for 24 h after	Evidence of muscle injury by MRI	↓ RT and LT posterior and medial
				CRP, hsCRP, ERS, MCP-1, FRAP, CAT, GPx, CK	† CRP, hsCRP, ERS, MCP-1, FRAP, CAT, GPx, CK
				IL-8	↓ IL-8
				Intensity of pain	† Intensity of pain
Sciberras et al., 2015 [4]	Double-blind randomized cross-over. Subjects performed three trials in total (supplement/placebo and control)	11 male recreational athletes (35.5 ± 5.7 years)	500 mg of curcumin in capsules (Meriva Curcumin) 72 h before and immediately before exercise	RPE	† RPE
				Cortisol, PCR, Hto, Hb,	† Cortisol, PCR, Hto, Hb,
				WBC, Neutrophils, IL-6,	WBC, Neutrophils, IL-6,
				IL1-RA, IL-10	IL1-RA, IL-10
				Questionnaire DALDA	↑ “better than usual”
Nicol et al., 2015 [28]	Double-blind crossover randomized controlled trial	17 moderately active men (33.8 ± 5.4 years)	2.5 g of curcumin in capsules, 48 h before the exercise and for 72 h after	Muscle pain—VAS	↓ Muscle pain: squatting jump (1.5−1.1; ± 1.2); Stretch butt (−1.0a-1.9; ± 0.9); Sitting on one leg (−1.4a−1.7; 90% CL ± 1.0)
				Jump height to one leg	↑ Jump 1 leg (15%; 90% CL ± 1 2%)
				CK	↓ CK 24 and 48 h before (−22%; 90% CL ± 22%), (−29%; ± 21%)
				IL-6	↑ IL-6 at 0-h (31%; ± 29%) and 48 h (32%; ± 29%) ↓ 24 h post-exercise (−20%; ± 18%)
				TNF-α	† TNF-α
Tanabe et al., 2015 [2]	Single-blind crossover randomized controlled trial	14 young men without regular resistance training (23.5 ± 2.3 years)	150 mg curcumin capsules (Theracurcumin Theravalues) twice a day 1 h before exercise and 12 h later	MVC Torque	↓ MVC Torque
				ROM	† ROM
				Upper arm circumference	† Upper arm circumference
				Muscle pain—VAS	†VAS
				CK	↓ CK (maximum activity)
				IL-6	† IL-6
				TNF-α	† TNF-α
**B. Summary of Studies Included in This Systematic Review (Continued)**					
**Author/s—Year**	**Study Design**	**Population**	**Intervention**	**Analyzed Results**	**Main Conclusions**
McFarlin et al., 2016 [30]	Randomized controlled trial double blind	28 men and women without regular resistance training (20 ± 1 ages and 19 ± 2 ages)	400mg curcumin capsules (Long-life) 48h before exercise and for 72h after	Subjective quadriceps pain	† Subjective quadriceps pain
				ADL	†ADL
				CK	↓CK
				TNF-α	↓TNF-α
				IL-6	†IL-6
				IL-8	↓IL-8
				IL-10	†IL-10
Nakhostin-Roohi et al., 2016 [24]	Controlled test randomized crossed double-blind	10 young men without regular training with weights (25.0 ± 1.6 years)	150 mg of curcumin gin capsules (Theravalues) Immediately after exercise	Muscle pain—VAS	↓ VAS 48–72 h
				TAC	↑ TAC
				CK	↓ CK
				ALT	↓ ALT
				AST	↓ AST
Delecroix et al., 2017 [31]	A randomized, balanced cross-over	10 rugby players elite level (20.7 ± 1.4 years)	2 g curcumin + 20 mg of piperine in capsules (MGD Nature) 3 times/day48h before exercise and for 48 h after exercise	6-s power sprint	< Group reduction EXP: (−1.77 ± 7.25%; 1277 ± 153 W). CON Group (−13.6 ± 13.0%; 1130 ± 241 W)
				CMJ	↑ CMJ (ES = −0.56; CI 90% = 0.81−0.32)
				CK	† CK 24, 48, 72 h post-exercise
				Muscle pain—Hooper scale	† Muscle pain—Hooper scale
				Subjective quadriceps pain	† Subjective quadriceps pain
Tanabe et al., 2018 [1]	Double-blind crossover randomized controlled trial	Exp1: 10 men (28.5 ± 3.4 years) Exp2: 10 men (29.0 ± 3.9 years) Both untrained 3-7 days prior to assay	90mg curcumin capsules (Theracurcumin Theravalues) 2 times/day Exp1: 7 days before exercise Exp2: 7 days after exercise	MVC Torque	Exp1:† Exp2:↑ MVC Torque
				ROM	Exp1:† Exp2:↑ ROM
				Muscle pain -VAS	Exp1:† Exp2: ↓ VAS
				T_2_	Exp1:† Exp2: † T_2_
				CK	Exp1:† Exp2:↓ CK
				TNF- α	Exp1:† Exp2: † TNF- α
				IL-8	Exp1: ↓ Exp2: † IL-8
				d-ROMs	Exp1:† Exp2: † d-ROMs
				BAP	Exp1:† Exp2: † BAP
Tanabe et al., 2019 [25]	Single-Blind Parallel Randomized Trial	24 young men without intense training during the study period PRE (28.8 ± 3.6 years) POST (29.8 ± 3.4 years) CON (28.0 ± 3.2 years)	90 mg curcumin capsules (Theracurcumin Theravalues) twice a day PRE: 7 days before exercise POST: 4 days after exercise CON: 4 days after exercise	MVC Torque	PRE: † POST: † MVC Torque
				ROM	PRE: † POST: ↑ ROM
				Muscle pain—VAS	PRE: † POST: ↓ VAS
				CK	PRE: † POST: † CK
**C. Summary of Studies Included in This Systematic Review (Continued)**					
**Author/s—Year**	**Study Design**	**Population**	**Intervention**	**Analyzed Results**	**Main Conclusions**
Jäger et al., 2019 [10]	Randomized controlled trial double-blind	63 men (31) and women (32) (21 ± 2 years) physically active meeting ACSM guidelines	G1: Placebo (PLB) G2: 50 mg curcumin in capsules (CurcuWIN^®^)G3: 200 mg curcumin in capsules (CurcuWIN^®^)3 times/day (breakfast/lunch/dinner)	Subjective muscle pain anterior, posterior, and total scale 100 mm Maximum extension torque and isokinetic flexion Extension power and isokinetic flexion Isometric torque Measurements: 1 h, 24 h, 48 h, and 72 h post-exercise	↑ Subjective muscle pain (anterior, posterior) G 1, 2, and 3 † Subjective (total) muscle pain G3 1 h and 24 h post-exercise † Maximum bending torque G2 † Bending power G2
Basham et al., 2019 [32]	Randomized controlled trial double-blind	20 men elite level (21.7 ± 2.9 years) physically active compliance with ACSM guidelines	1.5 g curcumin/69 mg curcuminoids 500 mg capsule (CurcuFresh, NOW FoodsUSA) twice a day (2 breakfast/1 dinner)	Oxidative stress Inflammation Muscle damage Muscle pain	↓ CK (*p* < 0.0001) ↓ VAS (*p* = 0.012) &TAC &MDA &TNF-α

(A) ↑: Statistically significant increase; †: change without statistical significance; ↓: Statistically significant decrease; MRI: magnetic resonance imaging; RT: right thigh; LT: left thigh; CRP: C-reactive protein; hsCRP: high sensitivity CRP; ERS: erythrocyte sedimentation; MCP-1: monocyte 1 chemotherapeutic protein; FRAP: Ferric reduction capacity of plasma; CAT: catalase; GPx: glutathione peroxidase; CK: creatine kinase; IL-8: interleukin 8; RPE: subjective perception of effort; Hto: hematocrit; Hb: hemoglobin; WBC: white blood cell count; IL-6: interleukin 6; IL1-RA: interleukin 1-RA; IL-10: interleukin 10; DALDA: daily analysis of life demands in athletes; VAS: visual analog scale; TNF-α: tumor necrosis factor alpha; CVS: maximum voluntary contraction; ROM: range of motion; TAC: Total Antioxidant Capacity; ALT: alanine aminotransferase; AST: aspartate aminotransferase. (B) ADL: Activities of daily living; CMJ: Contra movement jump; MVC Torque: Maximum voluntary contraction torque; ROM: Range of motion; T_2_: Transverse relaxation time; d-ROMs: Derivatives of reactive oxygen metabolites; BAP: Biological antioxidant potential; EXP: Experimental; CON: Control. (C) &: Unchanged; PLB: Placebo; G: Group; MDA: Malondialdehyde; ACSM: The American College of Sports Medicine.

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
