# Peer review of "Modulation of Exercise-Induced Muscle Damage, Inflammation, and Oxidative Markers by Curcumin Supplementation in a Physically Active Population: A Systematic Review"

_nutrients, 2020, doi:10.3390/nu12020501_

Round 1

Reviewer 1 Report

General Comment: This is an interesting review which provides information on nutraceutical potential of curcumin. However, as is mentioned in your “Limitations”, a low number of studies investigated in this subject and these studies are not homogenous.

Comments:

Page 1; Line 62: you should write “Natural” and not “natural” because you start a sentence.

Page 1; Line 63: after “bis” a gap is needed.

Page 1; Line 65: you should write “curcumin is……with multiple anti-inflammatory….”, and not “inflammatory”.

Page 1; Line 92: you should write “similar effects”

Page 5; Table 1: Sciberras et al., 2015 is the Reference 4 and not the 21. Additionally, in the last column T is referred to what? In page 6 (T) Total items fulfilled is referred to the T on the bottom!

Page 12: Line 257: it is necessary to write “hepcidin protein”, it is known that hepcidin is a protein.

Page 13; Line 283: you should write 2.5g and not 2,5g.

Page 13; Line 284: You should write 48h and 72h and not 48hs and 72hs. Additionally, what is exactly this VAS??? At the end of this sentence?

Page 15; Line 409: there is a small sentence “TNF-α also.” without any sense. Please rephrase the entire paragraph.

Page 16; 1st Paragraph: You should take into account that IL-6 could be a myokine in the case of exercise, and not really a pro-inflammatory cytokine.

Page 16; 2nd Paragraph: you should rephrase this paragraph.

Page 16; Line 458: write “malondialdehyde” and not “malonildialdeido”

Pay attention because in your manuscript symbols like kappa (k) in the NF-kB should be written by the same way. Normally, you should write with the Greek letter kappa (κ).

Author Response

Reviewer 1

Point-by-Point Response to Reviewer’s Comments

We would like to sincerely thank the reviewer for his/her helpful recommendations again. We have seriously considered all the comments and carefully revised the manuscript accordingly. Revisions are highlighted in yellow through the manuscript to indicate where changes have taken place. We feel that the quality of the manuscript has been significantly improved with these modifications and improvements based on the reviewers’ suggestions and comments. We hope our revision will lead to an acceptance of our manuscript for publication Nutrients.

Also, the manuscript has undergone English language editing by MDPI. The text has been checked for correct use of grammar and common technical terms, and edited to a level suitable for reporting research in a scholarly journal.

In advance,

King regards

Rev: Page 1; Line 62: you should write “Natural” and not “natural” because you start a sentence.

Authors: Thank you for your observation. Authors have included Natural instead natural in the text: “A natural product that can be used, with potential antioxidant and anti-inflammatory effects.”

Rev: Page 1; Line 63: after “bis” a gap is needed.

Authors: Thank you for your observation. Authors have added gap after bis: “(1,7-bis (4-hydroxy-3-methoxyphenyl) 1,6-heptadiene-3,5-dione)”

Rev: Page 1; Line 65: you should write “curcumin is……with multiple anti-inflammatory….”, and not “inflammatory”.

Authors: Thank you for your recommendation. Authors have written anti-inflammatory instead inflammatory in the sentence: “Curcumin is a highly pleiotropic molecule that interacts with multiple anti-inflammatory and antioxidant pathways [10, 11]”.

Rev: Page 1; Line 92: you should write “similar effects”

Authors: Thank you for your recommendation. Authors have added similar effects in that sentence: “Furthermore, similar effects of curcumin have been described by Tanabe et al. [1]”

Rev: Page 5; Table 1: Sciberras et al., 2015 is the Reference 4 and not the 21. Additionally, in the last column T is referred to what? In page 6 (T) Total items fulfilled is referred to the T on the bottom!

Authors: Thank you for your observation. Authors have changed 4 by 21: Sciberras et al. [4]. On the other hand, authors have included in table 1 footer the mean of TI and TF: “(TS) Total items fulfilled by study.  Last column (TI): Total items fulfilled by items”

Rev: Page 12: Line 257: it is necessary to write “hepcidin protein”, it is known that hepcidin is a protein.

Authors: Thank you for your recommendation. Authors have re-written that sentence: “elevation of hepcidin, which could be another potential cause of decreased iron level [38].”

Rev: Page 13; Line 283: you should write 2.5g and not 2,5g.

Authors: Thank you for your observation. Authors have changed 2.5g instead 2,5g in the text:that intake of 2.5g of curcumin supplementation”

Rev: Page 13; Line 284: You should write 48h and 72h and not 48hs and 72hs. Additionally, what is exactly this VAS??? At the end of this sentence?

Authors: Thank you for your recommendation. Authors have written 48h and 72h and not 48hs and 72hs in the texts: “supplementation in capsules taken 48h before and for 72h.” On the other hand, authors have deleted VAS.

Rev: Page 15; Line 409: there is a small sentence “TNF-α also.” without any sense. Please rephrase the entire paragraph.

Authors: Thank you for your observation. The sentence “TNF-α also” is a typo that has been removed.

Rev: Page 16; 1st Paragraph: You should take into account that IL-6 could be a myokine in the case of exercise, and not really a pro-inflammatory cytokine.

Authors: Thank you for your observation. Authors haver re-written that sentence:On the other hand, curcumin supplementation showed a downward but non-significant trend on IL-6 cytokine derived from exercise practice [2, 4, 30].”

Rev: Page 16; 2nd Paragraph: you should rephrase this paragraph.

Authors: Thank you for your recommendation. Authors have rephrased that paragraph: “In this way, the promoters of IL-6 and IL-8 cytokines possess binding sites for NF-κB, C/EBPβ, and c-Jun [49]. We believe the role of NF-κB inhibition is a therapeutic objective of curcumin in inflammation because of the importance of NF-κB for the regulation of the constitution and expression of IL-6 and IL-8 [49, 50]. McFarling et al. [30] concretely observed that significant inhibition of NF-κB could be related to a significant decrease in IL-8 and a downward trend in IL-6. Thus, supplementation with 400 mg of curcumin, two days before and three days after exercise, appears to be effective in attenuating exercise-induced inflammation because of its direct action on NF-κB which influences the cytokines IL-8 and IL-6”

Rev: Page 16; Line 458: write “malondialdehyde” and not “malonildialdeido”

Authors: Thank you for your recommendation. Authors have changed “malondialdehyde” and not “malonildialdeido”: “… and malondialdehyde is formed by lipid peroxidation”

Rev: Pay attention because in your manuscript symbols like kappa (k) in the NF-kB should be written by the same way. Normally, you should write with the Greek letter kappa (κ).

Authors: Thank you for your observation. All errors were corrected as nuclear factor-κB (NF-κB).

Reviewer 2 Report

To begin with, there are abundant careless mistakes.  

The following are found only from the abstract, introduction parts.

Line 24: …has been shown to have mainly antioxidant and also antiinflammation… Revision required here.

Line 37: CK, full term should be provided at its first-time presence.

Line 37: increase muscle performance…. Also curcumin have….

Increases, and has

Line 38: …pro-inflammatory cytokines as TNF-α, IL-6 and IL-8

such as…

Line 39: …supplementation with 150 and 1500 mg/day of oral curcumin before and until 72 hours after exercise has been shown to be effective on exercise performance modulated in part by reduction on EIMD and inflammation caused by physical activity.

Line 92: Also, Similar effects… Similar? Capital letter?

Line 100-102: Incomplete sentence, difficult to understand.

Line 103: However, is necessary for clarification the useful doses… Missing “it” in front of is.

Further, the toxicity of supplementary curcumin should be summarized in the introduction, which will help the readers build a cautious concept on dietary curcumin consumption, especially the pulmonic and hepatic damages.

The methodology is beautiful and the inclusion / exclusion criteria are well defined.

Only one minor concern here, why the authors include “6 g de curcumin + 60mg de piperine / day” study from Ref31? How the authors verify the health-promoting effects only coming from curcumin not piperine?

The authors also made a great job in Discussions part. They covered all the data and mechanisms involved.

Also, the bioavailability and toxicity of curcumin should be discussed in more details in  Limitations and Practical applications parts, respectively.

Author Response

Reviewer 2

Point-by-Point Response to Reviewer’s Comments

We would like to sincerely thank the reviewer for his/her helpful recommendations again. We have seriously considered all the comments and carefully revised the manuscript accordingly. Revisions are highlighted in green through the manuscript to indicate where changes have taken place. We feel that the quality of the manuscript has been significantly improved with these modifications and improvements based on the reviewers’ suggestions and comments. We hope our revision will lead to an acceptance of our manuscript for publication Nutrients.

Also, the manuscript has undergone English language editing by MDPI. The text has been checked for correct use of grammar and common technical terms, and edited to a level suitable for reporting research in a scholarly journal.

In advance,

King regards

Rev: Line 24: …has been shown to have mainly antioxidant and also antiinflammation… Revision required here.

Authors: Thank you for your recommendation. Authors have written anti-inflammatory instead inflammatory in the sentence: “Curcumin is a highly pleiotropic molecule that interacts with multiple anti-inflammatory and antioxidant pathways”

Rev: Line 37: CK, full term should be provided at its first-time presence.

Authors: for decrease of creatine kinase (CK)

Rev: Line 37: increase muscle performance…. Also curcumin have…. Increases, and has

Authors: Thank you for your recommendation. Authors have added creatine kinase in that sentence: “Curcumin is a highly pleiotropic molecule that interacts with multiple anti-inflammatory and antioxidant pathways [10, 11]”.

Rev: Line 38: …pro-inflammatory cytokines as TNF-α, IL-6 and IL-8

such as…

Authors: Thank you for your recommendation. Authors have added such in that sentence: “cytokines such as TNF-α, IL-6 and IL-8”

Rev: Line 39: …supplementation with 150 and 1500 mg/day of oral curcumin before and until 72 hours after exercise has been shown to be effective on exercise performance modulated in part by reduction on EIMD and inflammation caused by physical activity.

Authors: Thank you for your recommendation. Authors have rephrased that paragraph:

“In summary, the administration of curcumin at a dose between 150-1500 mg/day before and during exercise, and up until 72 hours’ post-exercise, improved performance by reducing EIMD and modulating the inflammation caused by physical activity. In addition, humans appear to be able to tolerate high doses of curcumin without significant side-effects.”

Rev: Line 92: Also, Similar effects… Similar? Capital letter?

Authors: Thank you for your observation. The sentence “similar” is a typo that has been removed. “Also, similar effects of”

Rev: Line 100-102: Incomplete sentence, difficult to understand.

Authors: Thank you for your recommendation. Authors have rephrased that paragraph: “Finally, not all observed changes in performance and soreness after exercise in humans [10] have been reproduced on the mouse model [26]”.

Rev: Line 103: However, is necessary for clarification the useful doses… Missing “it” in front of is.

Authors: Thank you for your observation. Authors have included it instead natural in the text: “However, it is necessary for clarification”

Rev: Further, the toxicity of supplementary curcumin should be summarized in the introduction, which will help the readers build a cautious concept on dietary curcumin consumption, especially the pulmonic and hepatic damages.

Authors: Thank you for your recommendation. Authors have added this paragraph:

“In this way, curcumin used as a pharmaceutical preparation has been shown to be safe, even at high doses. However, it has been shown to cause some gastric irritation in humans, hepatotoxicity in mice, and at high doses, hepatotoxicity in rats. Humans appear to be able to tolerate high doses of curcumin without significant side-effects. This may be because of differences in metabolism of curcumin in humans as compared to susceptible species such as rats. However, when used as a spice, because of its high water content, it could be attacked by aflatoxin-producing fungi, causing kidney, lung, or liver toxicity. It must be taken into account that curcumin as a spice is produced in tropical countries (warm and humid) that favor the growth of fungi [11, 12].”

Rev: The methodology is beautiful and the inclusion / exclusion criteria are well defined.

Authors: Thank you very much for your comment.

Rev: Only one minor concern here, why the authors include “6 g de curcumin + 60mg de piperine / day” study from Ref31? How the authors verify the health-promoting effects only coming from curcumin not piperine?

Authors: Thank you very much for your comment. Your question could be answered

Piperine is a thermonutrient that exerts its thermogenic action on the epithelial cells of the small intestine, increasing the rate of nutrient absorption and therefore increasing its bioavailability. Piperine has been administered together with vitamins, minerals and some nutrients as a bioavailability enhancer and is not genotoxic and did not present any significant adverse effects at supraphysiological doses (5-20 times higher than normal human intake) on internal organs, weight, hemoglobin levels, total serum proteins, albumin, cholesterol, fats and nitrogen. Patil UK, Singh A, Chakraborty AK. Role of piperine as a bioavailability enhancer. Inter J Recent Advan Pharma Resear. 2011 Apr;4:16-23.

Because of our experience in using piperine, recently in professional rowing athletes we have evaluated the comparative efficacy of two oral Fe supplements, one supplement with a dose of 325mg in the form of ferrous sulfate and another with a dose of 50mg in the presence of BioPerine® during the 10-week sports season. Both supplements showed significant improvement in ferritin compared to the control group, but not in sports performancer. Fernandez-Lazaro D, Alfredo Córdova, Fernández-Lázaro CI, Alberto C, Juan M. Comparative efficacy of two iron oral supplements on hematological profile and sports performance in professional rowers: Evaluation of a BioPerine® Bioavailability Enhancer.  Ibero-American Congress on Nutrition; Pamplona: Rev Esp Nutr Hum Diet. 2019 June; 75 (S2): 174-75

Rev: The authors also made a great job in Discussions part. They covered all the data and mechanisms involved.

Authors: Thank you very much for your comment.

Rev: Also, the bioavailability and toxicity of curcumin should be discussed in more details in Limitations and Practical applications parts, respectively.

Authors: Thank you for your observation. Authors have included a paragraph in limitations in the manuscript: “Another way to improve bioavailability would be through piperine, as demonstrated by Delecroix et al. [31], who included 6 g of curcumin + 60 mg of piperine/day. Piperine is a thermonutrient that exerts its thermogenic action on the epithelial cells of the small intestine, increasing the rate of nutrient absorption and therefore increasing its bioavailability [54].”Patil UK, Singh A, Chakraborty AK. Role of piperine as a bioavailability enhancer. Inter J Recent Advan Pharma Resear. 2011 Apr;4:16-23

Thank you for your observation. Authors have included a paragraph in practical applications in the manuscript: “Finally, curcumin should always be used as a pharmaceutical preparation and not as a spice to avoid the toxic effects of fungical aphlotoxins. In addition, caution is needed with athletes who are sensitive to gastric irritation”.

Round 2

Reviewer 1 Report

Since the authors took into account Reviewer’ comments, I think that the new revision should be published.

The only minor comment is:

All new paragraphs should be written using the same paragraph’s style as the entire manuscript.